# Adaptive GNN for Image Analysis and Editing

**Lingyu Liang**
South China Univ. of Tech.
lianglysky@gmail.com

**Lianwen Jin**[*]
South China Univ. of Tech.
lianwen.jin@gmail.com

**Yong Xu**[*]
South China Univ. of Tech.
Peng Cheng Laboratory
yxu@scut.edu.cn

## Abstract

Graph neural network (GNN) has powerful representation ability, but optimal configurations of GNN are non-trivial to obtain due to diversity of graph structure and cascaded nonlinearities. This paper aims to understand some properties of GNN from a computer vision (CV) perspective. In mathematical analysis, we propose an adaptive GNN model by recursive definition, and derive its relation with two basic operations in CV: filtering and propagation operations. The proposed GNN model is formulated as a label propagation system with guided map, graph Laplacian and node weight. It reveals that 1) the guided map and node weight determine whether a GNN leads to filtering or propagation diffusion, and 2) the kernel of graph Laplacian controls diffusion pattern. In practical verification, we design a new regularization structure with guided feature to produce GNN-based filtering and propagation diffusion to tackle the ill-posed inverse problems of quotient image analysis (QIA), which recovers the reflectance ratio as a signature for image analysis or adjustment. A flexible QIA-GNN framework is constructed to achieve various image-based editing tasks, like face illumination synthesis and low-light image enhancement. Experiments show the effectiveness of the QIA-GNN, and provide new insights of GNN for image analysis and editing.

## 1 Introduction

Recently, many research efforts have been devoted to graph neural network (GNN) [1–3], which is a significant deep learning technique for graph data under semi-supervised learning. Despite its powerful representation ability, optimal configurations of GNN are not trivial to obtain due to diversity of graph structure and cascaded nonlinearities. The layer structure or parameters are mostly determined by experimentations with expertise. In this paper, we intend to understand some properties of GNN mathematically from a computer vision (CV) perspective, and develop some GNN-based operations for CV problems.

In image analysis and synthesis, there are two basic operations. One is ***filtering*** to suppress or extract feature/content in images [4–7]; the other is ***propagation*** that diffuses the visual feature from the representative region throughout the entire image, so that similar pixels/regions have similar visual appearance [8–12]. The reviving of neural networks with deep learning has introduced many CNN-based networks to achieve filtering or propagation [6, 5, 13–16]. However, the properties of these models have not been clearly understood, since they found their inspirations in diverse

---

[*]**Corresponding authors: Lianwen Jin, Yong Xu.** Lingyu Liang and Lianwen Jin are supported by Natural Science Foundation of Guangdong Province (No. 2017A030312006, 2019A1515011045), the National Key Research and Development Program of China (No. 2016YFB1001405), NSFC (Grant No.: 61673182, 61771199, 61502176), GDSTP (No. 2017A010101027), GZSTP (No. 201704020134) and Fundamental Research Funds for the Central Universities (No. 2019MS023); Yong Xu is supported by National Nature Science Foundation of China (61672241, U1611461), Natural Science Foundation of Guangdong Province (2016A030308013), and Science and Technology Program of Guangzhou (201802010055).

contexts and were formulated in diverse forms, like partial differential equation (PDE) [8], variational functional [10, 11] or deep neural networks (DNN) [6, 16].

Recent research indicates that standard signal processing models with elaborate prior knowledge can achieve competitive performance with the state-of-the-art DNN-based methods [4, 9, 17, 8, 10]. It inspires us to analyze DNN based on existing visual operations. Specifically, we focus on GNN, which is one of the semi-supervised methods in the deep learning on graphs [1]. We try to explore different diffusion properties of GNN with its relation to filtering or propagation models.

In mathematical analysis, we propose a GNN model by recursive definition, which is formulated as a graph-based label propagation system with guided map, graph Laplacian and node weight. We derive its relation with the basic CV operations of filtering (e.g. edge-aware filter) and propagation (e.g. learning to diffuse (LTD) model). It reveals that: 1) Guided map and node weight determine whether a GNN leads to filtering or propagation diffusion; 2) Kernel of graph Laplacian controls diffusion pattern of a GNN.

In practical analysis, we applied the GNN models to design operations and systems for quotient image analysis (QIA), which recovers the reflectance ratio as a signature for illumination analysis or adjustment [18]. QIA is an essential component for image analysis and synthesis [19–30], especially for face synthesis and low-light image enhancement.

The challenges of QIA is twofold. Firstly, QIA extracts intrinsic representations from ambiguous and noisy data, which is an ill-posed inverse problem [31]. Secondly, slight errors in QIA may lead to obvious visual artifacts, since human visual cognitive systems are highly sensitive to image appearance changes.

To tackle these problems, we impose new adaptive kernel structures with different guided feature and priors to the GNN model, and propose GNN-based operations that achieve specific diffusion of filtering and propagation for QIA. Then, we develop a GNN-based QIA system (QIA-GNN) for image-based illumination synthesis. The system consists of three GNN subnetwork (denoted as QIA-GNN-L1/L2/L3). QIA-GNN-L1 acts as a filtering operation, which extracts the initial reflection and illumination feature from input images; QIA-GNN-L2 acts as a propagation operation, which adaptively propagates the initial illumination feature from the representative region to the whole images with good visual consistency; QIA-GNN-L3 is the output layer that combines different image layers and features to obtain the synthesized results. In this paper, we simply construct QIA-GNN-L1/L2 under semi-supervised learning scheme, but the GNN-based system is flexible to achieve various illumination synthesis tasks, including face relighting, face swapping, transfiguring and low-light enhancement.

The main contributions are summarized as follows:

- We propose an adaptive GNN model for image analysis, and mathematically derive its relation to filtering and propagation models, like edge-aware filters [32] and the LTD model [8]. It reveals that when a GNN is formulated as a graph-based label propagation system with guided map, graph Laplacian and node weight, then guided map and node weight determine whether it produces filtering or propagation diffusion; kernel of graph Laplacian controls its diffusion pattern.

- To tackle the inverse problems of QIA, we design a new adaptive kernel with different guided feature and priors, and propose GNN-based operations that achieve specific diffusion of filtering and propagation for QIA. Then, a flexible QIA-GNN system is constructed to produce various illumination synthesis, such as face relighting, face swapping, transfiguring and low-light enhancement.

## 2 Proposed GNN with Adaptive Kernel

The original GNN was proposed in "pre-deep-learning" era [33]. For a graph $\mathbb{G} = (\mathbb{V}, \mathbb{E})$ with $N$ nodes ($\mathbb{V} = \{v_1, ...v_N\}$), a GNN can be formulated as a recursive equation:

$$\mathbf{u}_i = \sum_{j \in \mathcal{N}(i)} \mathcal{F}\left(\mathbf{u}_i, \mathbf{u}_j, \mathbf{p}_i, \mathbf{p}_j, \mathbf{h}_{i,j}\right), \tag{1}$$

where $\mathbf{u}_i$ is the state of node $v_i$; $j \in \mathcal{N}(i)$ is the neighborhood set of node $v_i$; $\mathbf{p}$ and $\mathbf{h}$ denote features of nodes and edges respectively; and $\mathcal{F}$ is a parametric function.

The GNN model of [33] was originally designed for classification or regression problems under a supervised learning scheme. This paper further extends and explores GNN in two aspects. Firstly, we mathematically distinguish and analyze two intrinsic diffusion properties of GNN, i.e. filtering and propagation; then we use the GNN to unify many significant CV operations, as discussed in Sec. 2.1 and Sec. 2.2. Secondly, we generalize the formulation of Scarselli's GNN [33] from data classification/regression to visual data manipulation, where we propose a new kernel structure for QIA in Sec. 2.3 and a 3-layer QIA-GNN system to achieve multi-task illumination editing in Sec. 3.

Here, we propose an adaptive GNN based on the graph-based label propagation (LP) system [34]. Different from the original LP system for labelling nodes of a graph [35], we formulate the diffusion process from a visual diffusion perspective, and it can achieve both filtering and propagation diffusion.

Let $\mathcal{V}$ be the visual element domain of an image. The image is mapped into a graph $\mathbb{G} = (\mathbb{V}, \mathbb{E})$, where each node $v_i \in \mathbb{V}, i = 1, ..., N$ corresponds to the visual element of the image, and $\mathbf{p}_i$ is the feature of node $v_i$. Let $\mathbf{u}$ be the state of visual element defined over $\mathcal{V}$, i.e. $\mathbf{u} \in \mathbb{R}^N$. The graph-based LP system of the GNN can be reformulated as:

$$\mathbf{u}^{t+1} - \mathbf{u}^t = \mathbf{L}\mathbf{u}^t + \Lambda(\mathbf{g} - \mathbf{u}^t) \tag{2}$$

- $\mathbf{g}(\mathbf{p})$ is the guided map defined over $\mathcal{V}$, i.e. $\mathbf{g} \in \mathbb{R}^N$, which is used to guide the diffusion of GNN. The representative visual elements of the guided map is defined within $\mathcal{S}$, where $\mathcal{S}$ is a close subset of $\mathcal{V}$ with boundary $\partial \mathcal{S}$ for diffusion of propagation;

- $\Lambda(\mathbf{p}) = \mathrm{diag}(\lambda(\mathbf{p}_i))_{i=1,...,N} \in \mathbb{R}^{N \times N}$ with $\lambda \geq 0$ is the node weight, which determines the restricted region and the level of the guidance map $\mathbf{g}$ for $\mathbf{u}$;

- $\mathbf{L}(\mathbf{p}) \in \mathbb{R}^{N \times N}$ is the graph Laplacian controlling the local diffusion pattern of GNN, where the kernel function $k(\mathbf{p}_i, \mathbf{p}_j)$ measures the similarity of a node with its neighborhood set as follows:

$$\mathbf{L}_{ij} = \begin{cases} k(\mathbf{p}_i, \mathbf{p}_j), & j \in \mathcal{N}(i) \\ -\sum_{\mathbf{p}_j \in \mathcal{N}_{\mathbf{p}_i}} k(\mathbf{p}_i, \mathbf{p}_j) & i = j \\ 0 & \text{otherwise.} \end{cases} \tag{3}$$

Eq. 2 can be solved iteratively using Jacobi method. It can be also proved that this LP system with graph Laplacian of Eq. 3 converges to a unique solution based on the Banach fixed-point theorem [36].

Studies indicate that GNNs can achieve state-of-the-art performance in various tasks [37–40], but the design of new GNNs is mostly based on empirical heuristics and trial-and-error. Recently, [3] proposes a theoretical framework for analyzing the expressive power of GNNs based on Weisfeiler-Lehman (WL) graph isomorphism test [41], and validates the theory by experiments for graph-focused tasks. In the following sections, we analyse the diffusion properties of GNN from a CV perspective. The propagation and filtering properties of GNN guide us to construct new GNN-based operations and system for image analysis and synthesis.

## 2.1 Propagation Properties of Proposed GNN

With proper setting of $\{\mathbf{g}, \Lambda, \mathbf{L}\}$, the proposed GNN model (2) can produce propagation diffusion, such as Zhu's LP model [35] or Liu's learning to diffuse (LTD) models [8].

In propagation diffusion, the representative elements of $\mathbf{g}$ are within $\mathcal{S}$, where $\mathcal{S} \subset \mathcal{V}$ with boundary $\partial \mathcal{S}$. The value of the reaction weight $\Lambda_{ii} = \lambda(\mathbf{p}_i)$ is determined depending on $v_i \in \mathcal{S}$ or not. The GNN model identifies the representative visual element of $\mathbf{g}$ and propagate the value from $\mathcal{S}$ to $\mathcal{V}$. When Eq. (2) is stable, the GNN model becomes:

$$\mathbf{L}\mathbf{u} + \Lambda(\mathbf{g} - \mathbf{u}) = 0 \tag{4}$$

with

$$\begin{cases} \lambda(\mathbf{p}_i) \text{ is large}, \mathbf{g}_i = s_{\mathbf{p}_i}, & v_i \in \mathcal{S} \\ \lambda(\mathbf{p}_i) \text{ is small}, \mathbf{g}_i = \epsilon, & v_i \in \mathcal{V} \setminus \mathcal{S}, \end{cases} \tag{5}$$

where $s_{\mathbf{p}_i}$ is the value corresponding to a node $v_i$ with feature $\mathbf{p}_i$ in representative domain $\mathcal{S}$; $\epsilon$ is a small constant to avoid degeneration or $\epsilon = 0$. The specific value of $\lambda(\mathbf{p}_i)$ is task-dependent. One typical setting is $\lambda(\mathbf{p}_i) = 1$ for $\mathbf{p}_i \in \mathcal{S}$, while $\lambda(\mathbf{p}_i) = 0$ otherwise.

To clearly demonstrate the propagation properties of the GNN model, we reformulate the whole LP system (4) with setting (5) for each node $v_i$ with its neighborhood $j \in \mathcal{N}(i)$:

$$\left( \sum_{j \in \mathcal{N}(i)} k(\mathbf{p}_i, \mathbf{p}_j) + \lambda(\mathbf{p}_i) \right) \mathbf{u}_i - \sum_{j \in \mathcal{N}(i)} k(\mathbf{p}_i, \mathbf{q}_j) \mathbf{u}_j = \lambda(\mathbf{p}_i) \mathbf{g}_i$$

$$\Rightarrow \mathbf{u}_i = \frac{1}{d_{\mathbf{p}_i} + \lambda(\mathbf{p}_i)} \left( \sum_{j \in \mathcal{N}(i)} k(\mathbf{p}_i, \mathbf{p}_j) \mathbf{u}_j + \lambda(\mathbf{p}_i) \mathbf{g}_i \right) \tag{6}$$

where $d_{\mathbf{p}_i} = \sum_{j \in \mathcal{N}(i)} k(\mathbf{p}_i, \mathbf{p}_j)$. The value of $\mathbf{u}_i$ in (6) is mainly controlled by $\lambda(\mathbf{p}_i)$:

- For $v_i \in \mathcal{S}$, $\lambda(\mathbf{p}_i)$ is large, then $\mathbf{u}_i$ is dominated by the term of the guided map $\frac{\lambda(\mathbf{p}_i)}{d_{\mathbf{p}_i} + \lambda(\mathbf{p}_i)} \mathbf{g}_i$;
- For $v_i \in \mathcal{V} \setminus \mathcal{S}$, $\lambda(\mathbf{p}_i)$ is small, then $\mathbf{u}_i$ is determined by the diffusion of Eq. (6).

### 2.1.1 Relation to Zhu's Label Propagation (LP)

Let $\mathbf{u} = \mathbf{u}^{LP} = (\mathbf{u}_l^{LP}, \mathbf{u}_u^{LP})$ specifies how each data is to be labeled, where $\mathbf{u}_l^{LP}$ denotes labeled data for $v_i \in \mathcal{S}$ and $\mathbf{u}_u^{LP}$ denotes unlabeled data for $v_i \in \mathcal{V} \setminus \mathcal{S}$. With the setting of $\{\mathbf{g}^{LP}, \Lambda^{LP}, \mathbf{L}^{LP}\}$ below, the GNN leads to Zhu's label propagation [35, 42]:

- $\mathbf{g}^{LP} = (\mathbf{g}_l^{LP}, \mathbf{g}_u^{LP})$, where $\mathbf{g}_l^{LP}$ denotes the initial label for $v_i \in \mathcal{S}$, and $\mathbf{g}_u^{LP} = 0$ for unlabeled data $v_i \in \mathcal{V} \setminus \mathcal{S}$;
- $\Lambda^{LP} = \mathrm{diag}(\lambda^{LP}(\mathbf{p}_i))_{i=1,\dots,N}$, where

$$\begin{cases} \lambda^{LP}(\mathbf{p}_i) \gg d_{\mathbf{p}_i}^{LP}, & v_i \in \mathcal{S} \\ \lambda^{LP}(\mathbf{p}_i) = 0, & v_i \in \mathcal{V} \setminus \mathcal{S}; \end{cases}$$

- $\mathbf{L}^{LP}$ uses the Gaussian kernel of width $\sigma$ as the similarity measurement, where $k^{LP}(\mathbf{p}_i, \mathbf{p}_j) = e^{-\frac{\|\mathbf{p}_i - \mathbf{p}_j\|^2}{2\sigma^2}}$. For $\mathbf{L}^{LP}$, we make a decomposition as $\mathbf{L}^{LP} = \mathbf{D}^{LP} + \mathbf{W}^{LP}$, where $\mathbf{D}_{ii}^{LP} = d_{\mathbf{p}_i}^{LP}$ is the diagonal component of $\mathbf{L}^{LP}$, and $\mathbf{W}^{LP}$ is the off-diagonal component. Then, we obtain $\mathbf{D}_{ii}^{LP} = \sum_j \mathbf{W}_{ij}^{LP}$.

With these settings, we can reformulate Eq. (4) as:

$$\mathbf{u}_i^{LP} = \begin{cases} \mathbf{g}_l^{LP}, & v_i \in \mathcal{S} \\ \frac{1}{d_{\mathbf{p}_i}^{LP}} \sum_{j \in \mathcal{N}(i)} k^{LP}(\mathbf{p}_i, \mathbf{p}_j) \mathbf{u}_j, & v_i \in \mathcal{V} \setminus \mathcal{S} \end{cases} \tag{7}$$

which is precisely one iteration of the label propagation [35].

### 2.1.2 Relation to Liu's Learning to Diffusion (LTD) Model

Similarly, we could also derive the relation to Liu's LTD model [8]. For $v_i \in \mathcal{S}$, let $\lambda(\mathbf{p}_i) \gg d_{\mathbf{p}_i}$, then $\mathbf{u}_i \approx \frac{\lambda(\mathbf{p}_i)}{d_{\mathbf{p}_i} + \lambda(\mathbf{p}_i)} \mathbf{g}_i \approx \mathbf{g}_i = s_{\mathbf{p}_i}$. The GNN model (4) with setting (5) becomes:

$$\mathbf{u}_i \simeq \begin{cases} s_{\mathbf{p}_i}, & v_i \in \mathcal{S} \\ \frac{1}{d_{\mathbf{p}_i} + \lambda(\mathbf{p}_i)} \left( \sum_{j \in \mathcal{N}(i)} k(\mathbf{p}_i, \mathbf{p}_j) \mathbf{u}_j + \lambda(\mathbf{p}_i) \mathbf{g}_i \right), & v_i \in \mathcal{V} \setminus \mathcal{S} \end{cases} \tag{8}$$

If $k(\mathbf{p}_i, \mathbf{p}_j) = \exp(-\beta \|\mathbf{p}_i - \mathbf{p}_j\|^2)$, the GNN system of Eq. (8) leads to the LTD model [8].

## 2.2 Filtering Properties of Proposed GNN

With certain setting, the GNN model (2) leads to diffusion that is similar to edge-aware filter, like anisotropic diffusion [43] or optimization-based filter [44].

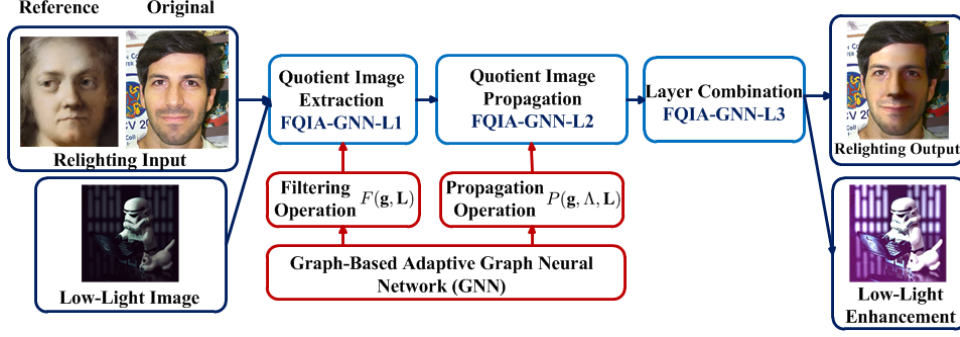

Figure 1: The QIA-GNN consists of three subnetworks: QIA-GNN-L1 acts as filtering operation for facial quotient image extraction; QIA-GNN-L2 acts as propagation operation for facial quotient image propagation; QIA-GNN-L3 produces the result that combines the image layers and feature.

In filtering diffusion, the representative elements of the guided map $\mathbf{g}$ cover the whole domain $\mathcal{S} = \mathcal{V}$ and reaction weight is a identity matrix $\Lambda = \mathbf{E}$. The GNN becomes:

$$\mathbf{u}^{t+1} - \mathbf{u}^t = \mathbf{L}\mathbf{u}^t + (\mathbf{g} - \mathbf{u}^t). \tag{9}$$

It can be regarded as the discrete form of bias anisotropic diffusion [45], if the kernel of $\mathbf{L}$ is controlled by the gradient of $\mathbf{u}$, i.e. $k = k(\|\nabla \mathbf{u}\|)$. Furthermore, if $\Lambda = 0$, we obtain the famous anisotropic diffusion [43].

### 2.2.1 Relation to Farbman's Optimization-Based Filter (OF)

Many important edge-aware filters [32] are defined implicitly by a variational formulation, and we called it optimization-based filters (OF) here. One of the representative OF model was proposed by [44], which is defined by the minimization of a quadratic functional:

$$\mathbf{u} = \underset{\mathbf{u}}{arg min.} \left\{ (\mathbf{u} - \mathbf{g})^\top (\mathbf{u} - \mathbf{g}) + \mathbf{u}^\top \mathbf{L}\mathbf{u} \right\}, \tag{10}$$

where $\mathbf{u}$ is the filtered output, $\mathbf{g}$ is the original image, and $\mathbf{L}$ encodes the filter kernel.

When Eq. (9) is stable, the GNN model becomes: $(\mathbf{E} - \mathbf{L})\mathbf{u} = \mathbf{g}$, which has the same solution of the quadratic functional of (10).

Let $\mathbf{u} = \mathbf{u}^{OF}$ be the filtered output, and $\mathbf{g} = \mathbf{g}^{OF}$ be the original image. With the setting below, the GNN (9) obtains edge-aware smoothing as OF of [44]:

- $\mathbf{L}^{OF}$ measures node similarity with $v_i$ and its 4-neighbor set $j \in \mathcal{N}_4(i)$ using the kernel:

$$k^{OF}(\mathbf{p}_i, \mathbf{p}_j) = \beta(\|\mathbf{p}_i^{OF} - \mathbf{p}_j^{OF}\|^\alpha + \varepsilon)^{-1}, \tag{11}$$

where $\mathbf{p}^{OF}$ is the log-luminance channel of $\mathbf{g}^{OF}$ to guide the edge-aware diffusion, $\alpha$ controls the local diffusion pattern, $\beta$ controls the global smoothness, and $\varepsilon$ is a small constant to avoid division by zero.

### 2.3 Adaptive Kernel for Quotient Image Analysis (QIA)

Based on the mathematical analysis, the diffusion pattern is controlled by the kernel of $\mathbf{L}$. To verify our analysis, we propose a new kernel with setting $\{d(\mathbf{M}), \mathbf{G}\}$ to design filtering and propagation operation for QIA, as shown bellow:

$$k_{QIA}(\mathbf{p}_i, \mathbf{p}_j) = \frac{d(\mathbf{M})_\mathbf{p}}{\|\mathbf{G}(\mathbf{p}_i) - \mathbf{G}(\mathbf{p}_j) + \epsilon\|_p^\alpha}, \tag{12}$$

where $d$ is a spatially inhomogeneous smoothness parameter to control the smoothness of propagation in different regions, which is determined by the confidence map $\mathbf{M}$; $\mathbf{M}$ can be obtained based on the structure information of an image, such as facial components or semantic segmentation of a scene; $\alpha$ controls the sensitivity of the term to the derivatives of the guided feature; $\mathbf{G}$ is the feature to guide the propagation, $\|\cdot\|_p$ represents the $p$-norm of guided feature space and $\epsilon$ is a small constant (typically $\epsilon = 0.001$) to avoid division by zero.

# 3 GNN for Quotient Image Analysis (QIA-GNN)

We apply the GNN with adaptive kernel to design adaptive filtering and propagation operations for QIA, and construct a GNN-based system (QIA-GNN) to achieve illumination-aware facial synthesis and low-light image enhancement, as shown in Fig. 1. QIA-GNN contains three subnetwork, denoted as QIA-GNN-L1/L2/L3:

1. QIA-GNN-L1: **Quotient image extraction**, where a GNN-based filtering operation $F(\mathbf{g}, \mathbf{L})$ is proposed to achieve two goals: 1) separating images into multiple facial layers; 2) extracting quotient image $\mathbf{Q}$ in the representative region.

2. QIA-GNN-L2: **Quotient image propagation**, where a GNN-based propagation operation $P(\mathbf{g}, \Lambda, \mathbf{L})$ is constructed that adaptively propagate $\mathbf{Q}$ to obtain illumination map $\mathbf{T}$. Note that different combination of $F(\mathbf{g}, \mathbf{L})$ and $P(\mathbf{g}, \Lambda, \mathbf{L})$ operations can produce different $\mathbf{T}$.

3. QIA-GNN-L3: **Image layer combination**, which combines $\mathbf{T}$ and the image layers to produce illumination editing.

We take *face relighting* as the main presentation in this paper, whose goal is to transfer the illumination from the reference image $\mathbf{R}$ to the input image $\mathbf{I}$ in a consistent manner. We construct the filtering $F^{relit}(\mathbf{g}, \mathbf{L})$ and propagation $P^{relit}(\mathbf{g}, \Lambda, \mathbf{L})$ operations of face relighting to show how to construct a GNN-based system with domain knowledge to solve the visual analysis problem.

## 3.1 Quotient Image Extraction (FQIA-GNN-L1)

QIA-GNN-L1 is constructed by the GNN-based filtering operation $F(\mathbf{g}, \mathbf{L})$ with facial prior, which separates the target $\mathbf{I}$ or reference $\mathbf{R}$ into facial layers and obtains the initial quotient image $\mathbf{Q}$. Note that some pre-processing, like landmark detection or face alignment, have been done for the input images. It is implemented as follow: Firstly, both the input $\mathbf{I}$ and $\mathbf{R}$ are converted into CIELAB color space, where the two chromaticity channels are regarded a color layers $\mathbf{I}_c$ ($\mathbf{R}_c$). Secondly, the luminance channel is decomposed into lighting layer $\mathbf{I}_L$ ($\mathbf{R}_L$) and detail layer $\mathbf{I}_d$ ($\mathbf{R}_d$) by $F(\mathbf{g}, \mathbf{L})$, where lighting layers captures the main illumination variance and detail layer contains facial details. Finally, the initial quotient image $\mathbf{Q}_{relit}$ is obtained by $\mathbf{Q}_{relit} = \frac{F^{relit}(\mathbf{R}_L|\mathbf{g},\mathbf{L})}{F^{relit}(\mathbf{I}_L|\mathbf{g},\mathbf{L})}$.

$F^{relit}$ acts as inhomogeneous filtering operation. To extract $\mathbf{Q}_{relit}$, $F^{relit}$ should smooth out details in background, eyes and eyebrows, while preserves the information in facial region. The setting of $F^{relit}(\mathbf{g}, \mathbf{L})$ is as follows:

- The guided map $\mathbf{g}$ is regarded as the input image to be filtered. For example, $\mathbf{g} = \mathbf{I}_L$ and $\mathbf{g} = \mathbf{R}_L$ lead to inhomogeneous smoothing of lighting layer $\mathbf{I}_L$ and $\mathbf{R}_L$, respectively.

- We integrate facial prior to the kernel $k_{QIA}$ of $\mathbf{L}$ to preserve the illumination within facial region, whiles smooth out the detail in eyes, eyebrows and background. Here we simply set $j \in \mathcal{N}_4(i)$ to obtain local filtering. $\mathbf{p} = log(\mathbf{I}_L)$ is the feature to guide the diffusion. Typically, the parameters are set as $\alpha = 1.2$ and $\varepsilon = 0.0001$. $d(\mathbf{M})$ is spatially determined by different region, so that background, eyes and eyebrows are smoothed out, while the informative illumination in the facial region is preserved.

## 3.2 Quotient Image Propagation (QIA-GNN-L2)

QIA-GNN-L2 is used to generate facial template $\mathbf{T}$ defined on $\mathcal{V}$ by propagating the values of $\mathbf{Q}$ from the facial region $\mathcal{S}$ to $\mathcal{V}$, i.e. $\mathbf{T}_{relit} = P^{relit}(\mathbf{Q}_{relit}|\mathbf{g}, \Lambda, \mathbf{L})$. Since human visual system correlates with the gradient in an image, $\mathbf{T}$ should fit the facial boundary closely and has the smooth transition between different regions.

To generate $\mathbf{T}_{relit}$ defined on $\mathcal{V}$, we construct $P^{relit}$ to propagate the information of $\mathbf{Q}_{relit}$ from the facial region $\mathcal{S}$ to the regions with missing and uncertain illumination, like eyes, eyebrows and background $\mathcal{V}\backslash\mathcal{S}$. The setting of $P^{relit}(\mathbf{g}, \Lambda, \mathbf{L})$ is as follows:

- For guided map, $\mathbf{g} = \mathbf{Q}_{relit}$, where $\mathbf{g}$ contains illumination of the quotient image in the representative region $\mathcal{S}$.

- The reaction weight $\Lambda$ determines which information is propagated to where. Therefore, the values of $\Lambda$ is consistent to the spatial location of $\mathcal{S}$ and $\mathcal{V}\backslash\mathcal{S}$ as $\Lambda = \text{diag}(\lambda_{relit}(\mathbf{p}_i))_{i=1,...,N}$, where

$$\lambda_{relit}(\mathbf{p}_i) = \left\{ \begin{array}{ll} 1, & v_i \in \mathcal{S} \\ 0, & v_i \in \mathcal{V}\backslash\mathcal{S}; \end{array} \right.$$

- For $k_{QIA}$, we produce the confidence map $\mathbf{M}$ that is consistent to $\mathcal{V}$ with smooth transition of region boundary. The visual information of $\mathbf{Q}_{relit}$ are propagated from the representative facial region $\mathcal{S}$ to the regions of eyes, eyebrows and background $\mathcal{V}\backslash\mathcal{S}$. Smoothness parameter $d$ are controlled by $\mathbf{M}$, so that $d$ is large (typically $d = 10$) in $\mathcal{V}\backslash\mathcal{S}$ to produce illumination propagation, and $d$ is small (typically $d = 0.4$) in $\mathcal{S}$ to preserve the significant illumination detail.

### 3.3 Image Layer Combination (QIA-GNN-L3)

QIA-GNN-L3 is output GNN layer, which combines $\mathbf{T}$ with the facial layer of original/reference to produce the final face synthesis. For face relighting, we transfer illumination of the reference to the original face by multiplying the $\mathbf{T}_{relit}$ and the lighting layer $\mathbf{I}_L$ as $\mathbf{O}_L = \mathbf{I}_L \circ \mathbf{T}_{relit}$, where $\circ$ is an element wise product. Finally, we recombine the other facial layers to obtain the face relighting output $\mathbf{O}$.

## 4 Experiment

### 4.1 Basic Evaluation

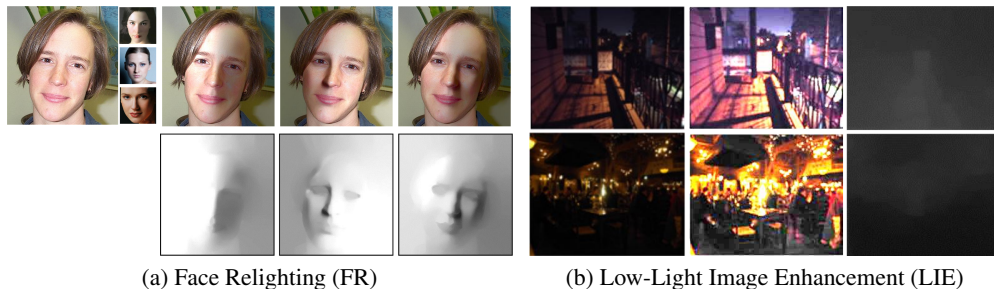

(a) Face Relighting (FR)            (b) Low-Light Image Enhancement (LIE)

Figure 2: Basic evaluation of QIA-GNN, where (a) shows face relighting with single target and multiple references; (b) shows low-light image enhancement with illumination maps.

We use the QIA-GNN to achieve face relighting (FR) and low-light image enhancement (LIE), as shown in Fig. 2. Fig. 2a shows FR of the same target with different references, and we can observe there is good consistency between illumination maps and the relighted results. Fig. 2b shows the LIE of different images, which indicates the effectiveness of our QIA-GNN system to capture the illumination feature in different scenes.

### 4.2 Qualitative Evaluation

We verify our QIA-GNN system for different editing tasks, including face relighting, face swapping, transfiguring and LIE. Fig. 3 illustrates the comparisons with the-state-of-arts, and indicates that the QIA-GNN system are competitive to related methods. Note that most of the previous systems are designed for specific tasks, while our GNN-QIA system is flexible to perform multiple image analysis or adjustment with the corresponding settings.

**Face Relighting (FR).** We compare our method with Li's [46] and Chen's [47] methods for face relighting. The results show that our method allows to relight faces in two patterns. For the first pattern, we perform QIA for all the RGB channels and obtain result similar to Li's method that transfers both the shading and tone to the target. For the other pattern, we perform QIA only for the

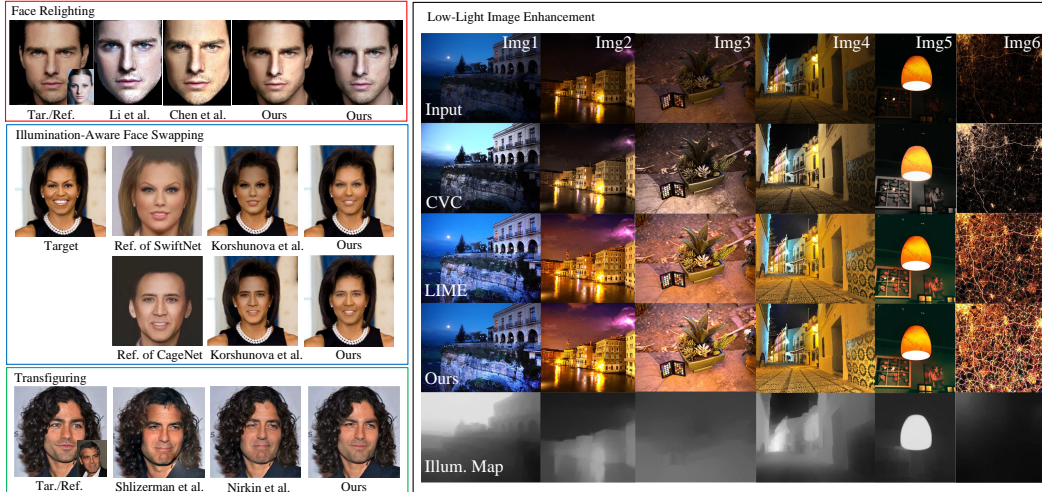

Figure 3: Qualitative comparisons with related methods, including **face relighting** (red box) with Li's [46] and Chen's [47], **face swapping** (blue box) with Korshunova's [20], **transfiguring** (green box) with Kemelmacher's [23] and Nirkin's [24], and **low-light image enhancement** (black box) with CVC [48] and LIME [28].

luminance channel of inputs, and obtain result that transfers only the shading of the reference but preserves the original tone of the target.

Furthermore, our method is complementary to previous methods in two aspects. For visual effects, [46] and [47] fail to relight the region outside the face, while ours adaptively generates the missing illumination in the background. For computation, [46] and [47] requires multiple operations derived from different contexts, while our operations are based on the same GNN model, which can be efficient to implement and extend.

**Illumination-Aware Face Swapping (FS).** Fig. 3 also shows the comparison with the recent works of [20] for face swapping. [20] proposed a new face synthesis system that trains a specific CNN to transform an input (original) identity into a reference identity with preserved facial properties. For example, the CageNet transforms the input identity into Nicolas Cage with the same expression. Although [20] has considered the lighting adjustment problem and integrates the lighting loss for the training of the CageNet, the shading and tone consistency could still be further improved by our method. Note that the QIA-GNN-L2 is setup to propagate quotient feature within the facial region for seamless blending, which is slightly different to the setting for relighting.

**Transfiguring (TF).** Recently, [23] introduced a new face synthesis task, called transfiguring, which let users transfigure their appearance from images by changing hair style, hair color etc. Fig. 3 shows the comparison of [23] and [24] for transfiguring. In some cases, some part of the faces is under occlusion of hair. To tackle this problem, we integrate the region-aware mask of [10] into our system and obtain competitive results compared with the state-of-the-art methods [23, 24]. Since the region-aware mask [10] is based on LP, it can be implemented by our GNN model, which indicates the powerful representation of GNN and the flexibility of our QIA-GNN system.

**Low-Light Image Enhancement (LIE).** Contrast enhancement have been extensively studied in recent decades [25, 49, 50, 27], but the enhancement for low-light images is still an unsolved problem [26, 51, 28–30, 52]. The main challenges is twofold. Firstly, the intensity of the images encodes many imaging factors, like illumination of the scene, reflection of the object, and the viewpoint. Obtaining good low-light enhancement without over-sharpening should recover or estimate some properties of the scene and object from image intensity [26, 51, 28–30, 52], but it is unfortunately an inherent ill-posed problem [53, 31]. Secondly, quality assessment of sharpened images in objective manners is still an open problem [54, 55], and it lacks a benchmark to evaluate the performance of different low-light enhancement methods. We focus on the first aspect in this paper, and apply the QIA-GNN with new regularization to adaptively enhance low-light image without over-sharpening.

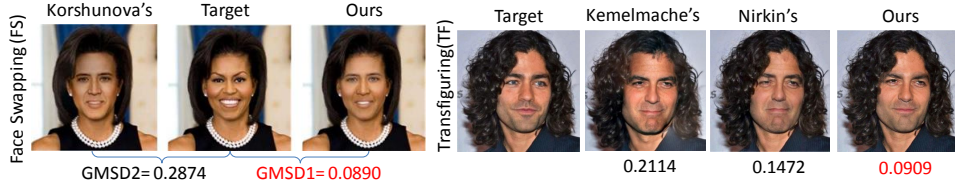

Figure 4: Objective assessment of FS with [20] and TF with [23] and [24] by GMSD [56].

|          | Img1 | Img2 | Img3 | Img4 | Img5 | Img6 |
|----------|------|------|------|------|------|------|
| CVC [48] | 6.54 | 6.28 | 6.38 | 6.75 | 4.27 | 5.16 |
| LIME [28]| **7.67** | 7.43 | 7.53 | **7.65** | 5.79 | 7.23 |
| Ours     | 7.48 | **7.47** | **7.55** | **7.65** | **6.04** | **7.71** |

Table 1: Objective assessment of LIE with CVC [48], LIME [28] and Ours by DE [57].

Based on the Retinex theory [53, 29], we obtain adaptive low-light enhancement via estimation and adjustment of the illumination maps of images using the QIA-GNN. It uses the QIA-GNN-L1 as filtering operation to extracts the initial illumination map, and adjusts the map adaptively with smooth transition by QIA-GNN-L2 which acts as propagation operation. Finally, the enhancement result is produced by combining the illumination map with the image layer.

We made comparison with Contextual and Variational Contrast enhancement (CVC) [48] and the recently proposed LIME [28] for low-lighting image enhancement, as shown in Fig. 3 (black box). The results indicates that our method facilitates to adaptively brighten images without over-sharpening the lighter regions of images in high dynamic range (HDR) manners. It also shows that our method outperforms CVC [48] with better tonal consistent and achieve competitive performance with the state-of-the-art LIME [28].

### 4.3 Quantitative Evaluation

For facial synthesis (FR, FS, TF), we made a small scale user study to determine which is more consistent to the original target with 10 volunteers (5 males and 5 females) for the results in Fig. 3, and our GNN-based results have a higher rank score than the other methods. A larger scale user study for more results would be performed in our future research. In addition, we used some metric of image quality assessment for objective evaluation. For FS and TF, we used gradient magnitude similarity deviation (GMSD) [56] to measure the visual similarity between the target and output pairs (shown in Fig. 4), where GMSD1<GMSD2 indicates that our method has better visual consistency than Korshunova's [20] for FS. Similarly, the results also indicate that our method (GMSD: 0.0909) is competitive to the Kemelmacher's [23] (GMSD: 0.2114) and Nirkin's [24] (GMSD: 0.1472), and obtains better visual consistency.

For low-light image enhancement, Table 1 shows the quantitative comparison between CVC [48], LIME [28] and ours by discrete entropy (DE) [57], where a higher value of DE indicates that the image has richer details. The objective measurements of Img 1 to Img 6 indicate that our method is superior to CVC [48] and competitive to the state-of-the-art LIME [28].

## 5 Conclusion

This paper proposes an adaptive GNN model by a LP system with guided map, graph Laplacian and node weight from CV perspective. We mathematically analyze its diffusion properties and derive its relation to edge-aware filter [32] and LTD model [8]. We find that different combination of the guided map and reaction weight determine whether a GNN leads to filtering or propagation diffusion, and the kernel of graph Laplacian controls the diffusion patterns. Based on the diffusion properties of GNN, we design a new adaptive kernel with different guided feature and image priors, and propose GNN-based operations that achieve specific diffusion of filtering and propagation for QIA. We also construct a QIA-GNN system, which is flexible to produce various image-based editing, such as face relighting, face swapping, transfiguring and low-light image enhancement. Experiments show the effectiveness of our methods and indicate that GNN can be a powerful tool for CV tasks.

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
