[Reviews · NeurIPS 2019]

Reviewer 1



* summary: This paper makes a connection between Graph neural network (GNN) and some computer vision tasks. They introduce an adaptive GNN formulated as a label propagation system, which can be related to two CV operations: filtering and propagation. Their adaptive GNN is designed based on guided map, graph Laplacian and node weight. The guided map and node weight are associated with filtering and propagation diffusion task in computer vision, and kernel of graph Laplacian is related to the diffusion pattern in computer vision task. They applied their model for quotient image analysis (QIA) and designed various illumination editing tasks for faces and scenes. * strengths: - The main idea of relating the GNN to some CV tasks is really interesting. - I like the way they narrate their work. After introducing their framework, they discuss how this model is a generalization of several models introduced for propagation diffusion and filtering. * Notes: - I think there was a lack of visualization or diagram in the paper. They could benefit from some pictures to make the main idea more understandable and easier to follow. - It would be very informative if they could explain the intuition that why and how “the guided map and node weight determines whether a GNN leads to filtering or propagation diffusion, and the kernel of graph Laplacian controls diffusion pattern.” - My main concern about the paper is the lack of enough experiments to show the efficacy of their propose model. They performed a qualitative experiments on a handful of images for the tasks of “face relighting,“ Illumination-Aware Face Swapping,“ “Transfiguring” as well as “low-light image enhancement.” They showed the results on a few images only. More importantly, there were not any qualitative experiments in the paper. How about post-hoc crowdsourced workers to rank the enhanced images; and then compare various methods together. - In the experiments, I would explain in a few sentences how their images are semi-supervised. - It would be very helpful if they could discuss the computational cost of their framework? - There is a minor grammatical mistake in sentence “199 in in background, eyes and eyebrows, while preserve the information in facial region. The setting of” - There was a question mark for Fig that needed to be corrected (I think it should be Fig 2). It is on page 7, last paragraph, second line.

Reviewer 2



The paper shows that the GNN model of Scarselli et al. (2009) can be adjusted (with many parameters) to be related to other filtering and diffusion methods that are based on a graph Laplacian $L$. However there is no significant and concrete novel insight that could be derived from these connections. At the end of Section 2, only obvious statement like "Based on the mathematical analysis, the diffusion pattern is controlled by the kernel of $L$" could be made. Important questions like "how to quantify the effectiveness of a kernel of $L$, and what is the best kernel of $L$ for a given task?" are not addressed. In fact, the kernel for QIA in Eq. (12) was chosen rather heuristically without any further analysis. Therefore, the paper does not really advance our understanding of GNN. The framework QIA-GNN in Section 3 was also constructed quite heuristically, and too complicated with three layers L1/L2/L3. There is no solid analysis and justification for these layers. The task QIA is subjective, with no clear objective metric for comparison. Therefore, it is difficult to assess the merit of the proposed GNN model. ***** POST REBUTTAL I have read the author response that provides some positive comparisons, both qualitative and quantitative. However, these experiments are small and could be cherry picked. I slightly increase my score from 4 to 5.

Reviewer 3



The paper addressed the quotient image based image editing with a single GNN framework. After describing the formulation, the experiment went on with the system with multiple different applications in image editing, including face relighting, swapping, transfiguring, image enhancement, etc. However, the paper are missing many experimental details. The most critical components of the experiments that are missing: (1) Are there any learning-based components in the system? If not, how much handcrafting is needed for different tasks? If yes, how are they trained? (2) What are the roles of each module (FQIA-GNN-L1, L2,L3 etc) in the various experiments? It might be helpful to show the illumination map along with the results. (3) What are the outputs of each module in each experiment? How are they obtained by these modules in different tasks? Some other questions/comments: (4) How does this method derive the two relighting results in Figure 2? What are the experimental settings that generate these two results? (5) There are weaknesses in the experiments. For example, in relighting, it would be beneficial to show relighting of multiple subjects with a single reference to demonstrate consistency. (6) On image enhancement, it would be better to show a quantitative evaluation, similar to the baseline [28].   In conclusion, although the paper is has an interesting and ambitious goal: a unified GNN framework for image editing, the delivery is not convincing. The experiments are weak and many details are missing. POST REBUTTAL comments: The authors seem to have tried to address most of my comments in the rebuttal regarding the level of details in the experiments and additional quantitative evaluations. The a uthors also included a small user study. I would increase the score from 4 to 6 after the rebuttal, and suggest that the authors include all additional experiments and visualizations in the final manuscript.

Reviewer 4



Originality: The paper proposes a novel and principled way of using and constructing GNNs for a wide variety of tasks. They claim their technique generalizes to multiple tasks. Clarity: The paper is fairly clear, well referenced, and well written. Some minor typos noted below. There are also some details in regards to the preprocessing of faces that could be clarified (see below). Significance: The paper proposes a principled way of constructing GNNs that appear to generalize to several useful computer vision applications. Additional experimental results and/or a small scale user study would be helpful to assess how the GNNs' output compare to other techniques and whether they generalize well. The authors should demonstrate their GNN is robust to both The main contribution here is providing some analysis on how these GNNs can be constructed in a principled way for this task. This way of parameterizing GNNs has the potential to be useful as parts of future computer vision pipeline. Typos: Line 78?: pre-deep-learning era Line 246?: generates Line 258: broken figure reference Line 273: We focus Line 283: rang -> range Consider adding a citation for "Retinex theory" line 275.

[Author Response · NeurIPS 2019]



Fig.1 Face relighting with GNN illum. maps      Fig.2 Lowlight image enhancement with GNN illum. maps

**Qualitative Experiments: 1)** We have made comparison with assorted editing tasks, including face relighting (FR), face swapping (FS), transfiguring (TF) and lowlight image enhancement (LIE). Note that all the related methods were designed for a specific task, while our GNN-based system is flexible to achieve multiple editing effects and obtain the state-of-the-art performance. Since many related methods have not released the source code, we choose some of typical results for comparison, and the experiments do illustrate the superiority of our GNN-based system. In fact, we have only shown parts of the results in this submission due to the page limitation, we would try our best to add more evaluation in the revised version, such as Fig.1-Fig.3.

**2)** Illumination maps are visualized: Fig.1 shows the FR of male/female with three different references, and the produced results and illumination maps demonstrate the consistency; Fig.2 shows the LIE of Img1 to Img6 (from left to right), which indicate the effectiveness of our GNN method to capture the illumination feature in different scenes.

|  | Img1 | Img2 | Img3 | Img4 | Img5 | Img6 |
|---|---|---|---|---|---|---|
| CVC | 6.54 | 6.28 | 6.38 | 6.75 | 4.27 | 5.16 |
| LIME | **7.67** | 7.43 | 7.53 | **7.65** | 5.79 | 7.23 |
| Ours | 7.48 | **7.47** | **7.55** | **7.65** | **6.04** | **7.71** |

Fig.3 Image quality assessment of FR and TF by GMSD.      Table1 Image quality assessment of LIE by discrete entropy.

**Quantitative Experiments: 1) For FR, FS and TF**, we made a small scale user study with 10 volunteers (5 males and 5 females) for the results in the submission, and the GNN results have a higher rank score than the other methods; a larger scale user study for more results would be performed in our journal paper. Furthermore, we used some metric of image quality assessment to for objective evaluation. In objective evaluation of FS and TF, we used gradient magnitude similarity deviation (GMSD) to measure the visual similarity between the target and output pairs (shown in Fig.3), where GMSD1<GMSD2 indicates that our method has better visual consistency than Korshunova's for FS. The results also indicate that our method is competitive to the Shlizerman's and Nirkin's, and obtains better visual consistency.

**2) For LIE**, Table1 shows the comparison between CVC, LIME and ours by discrete entropy (DE), where a higher value of DE indicates that the image has richer details. The objective comparison indicates that our method is superior to CVC and competitive to the state-of-the-art LIME.

**Originality: 1)** The GNN model of Scarselli et al. (2009) was originally designed for classification or regression under a supervised learning scheme. This paper further extend and explore GNN in two aspects. **Firstly**, we mathematically distinguish and analyze two intrinsic diffusion properties of GNN, i.e. filtering and propagation, and propose a GNN where guided map and node weight determine the diffusion type; the kernel of graph Laplacian controls the diffusion pattern. Then, we use the GNN to unify many significant CV operations from different fields, like Farbman's optimization-based filter and Liu's PDE-based LTD model. **Secondly**, we generalizes the formulation of Scarselli's GNN (2009) from data classification/regression to visual data manipulation. Since the diffusion type and pattern of GNN can be controlled by our framework, we propose a new kernel structure Eq.(12) with guided feature to construct filtering and propagation operations for QIA, and a three layer GNN-QIA system is built to achieve multi-task editing.

**2)** This paper focuses on the analysis of different diffusion type and pattern of GNN, and we unifies many significant filtering and propagation operations. Based on the work of this submission, more detailed theoretical analysis of the kernel $L$ would be studied in a much longer journal paper. In fact, the effectiveness of the kernel $L$ for a given task can also be analyzed and optimized via the illumination map, as shown in Fig.1, Fig.2, Fig.4.

**Technical Details:1)** Fig.4 visualizes workflow of GNN, where QIA-GNN-L1 acts as filtering operation to achieve quotient feature extraction, QIA-GNN-L2 acts as propagation operation to propagate and adjust the feature, QIA-GNN-L3 combines illumination map and image layers to obtain the final output.

Fig.4 Visualize the illum. map in each GNN-layer for relighting.

**2)** For FR in Fig.4, if we perform QIA only for the luminance channel of the inputs, we obtain the left output; if we perform QIA for all the RGB channels, we obtain the right output with color transfer.

[Meta-Review · NeurIPS 2019]

The paper proposes a new graph neural network architecture for vision tasks such as relighting and face swapping. By incorporating filtering and propagation mechanisms from classic graph Laplacian techniques, they recreate this behavior in deep neural networks. Through qualitative experiments, the paper argues that this leads to improved performance on the vision tasks considered. The reviewers generally agree that this is an interesting, novel approach, but have questions regarding evaluation. Since the results are qualitative, it is difficult to gain insight into general differences between the new method and previous ones. The reviewers encourage the additional results, such as the user study, presented in the authors' response to be added to the manuscript.